# Condylar Positional Changes in Skeletal Class II and Class III Malocclusions after Bimaxillary Orthognathic Surgery

**DOI:** 10.3390/jpm13111544

**Published:** 2023-10-27

**Authors:** Víctor Ravelo, Gabriela Olate, Marcio de Moraes, Claudio Huentequeo, Roberto Sacco, Sergio Olate

**Affiliations:** 1Division of Oral and Maxillofacial Surgery & CEMYQ, Universidad de La Frontera, Temuco 4780000, Chileclaudiohuentequeo@gmail.com (C.H.); 2PhD Program in Morphological Sciences, Facultad de Medicina, Universidad de La Frontera, Temuco 4780000, Chile; 3Division of Oral and Maxillofacial Surgery, Piracicaba Dental School, State University of Campinas, Piracicaba 13414-903, SP, Brazil; 4Division of Dentistry, Oral Surgery Department, School of Medical Sciences, The University of Manchester, Manchester M13 9PL, UK; 5Oral Surgery Department, King’s College Hospital NHS Trust, London SE5 9RW, UK

**Keywords:** temporomandibular joint, TMJ, facial deformity, orthognathic surgery, osteotomy

## Abstract

Orthognathic surgery is indicated to modify the position of the maxillomandibular structure; changes in the mandibular position after osteotomy can be related to changes in the position of the mandibular condyle in the articular fossa. The aim of this study was to determine changes produced in the mandibular condyle 6 months after orthognathic surgery. A cross-sectional study was conducted that included subjects who had undergone bimaxillary orthognathic surgery to treat dentofacial deformity of Angle class II (group CII) or Angle class III (group CIII). Standardized images were taken using cone-beam computed tomography 21 days before surgery and 6 months after surgery; measurement scales were used to identify the condylar position and its relations with the anterior, superior, and posterior joint spaces. The results were analyzed using the Shapiro–Wilk and Student’s *t*-tests, while considering a value of *p* < 0.05 as indicating a significant difference. Fifty-two joints from 26 patients, with an average age of 27.9 years (±10.81), were analyzed. All subjects in both group CII and group CIII showed a significant change in the anterior, superior, and posterior joint spaces. However, postoperative changes in the position of the condyle in the articular fossa were not significant in the anteroposterior analysis. We conclude that orthognathic surgery causes changes in the sagittal position of the mandibular condyle in subjects with mandibular retrognathism and prognathism.

## 1. Introduction

Cone-beam computed tomography (CBCT) is a tool used for diagnosis, surgical planning, and patient follow-up. For the temporomandibular joint, it allows precise observation of the bone structures of the joint [1], which improves the three-dimensional analysis of this joint by more accurately defining the condylar morphology and its potential alteration due to adaptive changes [2] or diseases.

It has been observed that male subjects have a larger condylar head volume than females [3], and among women, there is a greater prevalence of osteoarthrosis, which reduces the size of the mandibular condylar head, thereby causing a rupture between the adaptive capacity and the potential mechanical overload of the TMJ [4].

Dentofacial deformity is a morphological anomaly of the maxilla and mandible and is related to different levels of functional alterations. Subjects with dentofacial deformity with retrognathism or prognathism show sagittal maxillomandibular deficiency that affects function and esthetics [5,6]. Recent studies have shown that the size of the mandibular condyle is associated with the morphology of the mandible and the type of facial deformity, indicating that subjects with a smaller mandible have smaller condyles and subjects with facial deformity due to mandibular progeny have larger condyles [7].

Subjects with a dentofacial deformity are frequently treated with orthognathic surgery since the procedure is thought to enhance patients’ function and quality of life [8]. After surgery, it is possible to observe greater masticatory performance due to a stable dental occlusion, stability of the joint movement [9,10], and facial harmony and esthetics. 

The position of a mandibular condyle within the articular fossa can vary. Patients with a dentofacial deformity show a condylar position related to the morphological anatomy of the mandible [7]; in these cases, the morphology of the TMJ could be in an adaptation to optimize joint function in normality or joint disease [11]. It depends on variables such as age, sex, mechanical load conditions, and others. Thus, orthognathic surgery can modify the joint function and, consequently, the position of the condyle within the fossa.

The aim of this research was to determine the preoperative and postoperative positions of the mandibular condyle in patients who had undergone orthognathic surgery and to define the position change of the condyle in the articular fossa.

## 2. Materials and Methods

A cross-sectional study was conducted to analyze the position of the mandibular condyle of the temporomandibular joint (TMJ) in the sagittal and coronal views at the preoperative stage (T1) and after orthognathic surgery (T2). The subjects signed an informed consent, and this study was conducted in accordance with the Declaration of Helsinki. 

Fully dentate subjects over 18 years of age, both male and female, were included; all of them had a dentofacial deformity with Angle class II (group CII) and class III (group CIII) malocclusions. To determine the skeletal class, Steiner’s analysis was used to establish the angle A-N-B planes (point A: landmark located in the most anterior region of the anterior concavity of the maxilla; N: landmark in the most anterior point of the frontonasal suture; point B: landmark located in the most anterior region of the anterior concavity of the mandible). Subjects with an angle >4° were considered group CII, and those with an angle <0° were considered group CIII. Subjects with missing incisors, canines, bicuspid, and molars, except the third molars; subjects with previous orthognathic surgery, a history of facial trauma, or the presence of syndrome; and subjects with facial asymmetry with a chin deviation greater than 5 mm from the facial midline were excluded. 

The same surgical team performed all the surgeries. Each surgery was planned using a software with virtual planning and CAD-CAM for the surgical guides. The osteotomies were performed using a piezoelectric system (Satelec, Action, France), and in each case, the surgical sequence began with a mandibular surgery. In the maxilla, the Le Fort I (LFI) osteotomy was performed via intraoral access according to the routine technique; all cases were fixed with 4 osteosynthesis plates using the 2.0 system (Enterprises, Artfix Implants, Pinhais, PR, Brazil). For the mandible, the sagittal split ramus osteotomy (SSRO) was performed via the intraoral approach according to a previously published technique [12]. In all cases, the osteotomy was stabilized using 2.0 plates with monocortical screws (Enterprises, Artfix Implants, Pinhais, PR, Brazil); bicortical fixations were not used. The surgical movements were performed in relation to the requirements of each patient, including functional and aesthetic improvements. Preoperative imaging was performed within 21 days prior to the surgery, and the second image was obtained 6 months after the surgery. 

### 2.1. Image Analysis

A 3D image obtained via cone-beam computed tomography (CBCT) was used and analyzed using the NewTom 3D software, VGi EVO model (Verona, Italy), with a visualization field of 24 cm high and 19 cm width, and exposure parameters 110 kV, 8 mA, and 15 s. The image was obtained by a radiologist when the patient was in a vertical position while keeping the lips at rest without forcing the body position.

#### 2.1.1. Sagittal Measurements of Temporomandibular Joint (TMJ)

Anatomical landmarks of the condyle were used for sagittal and coronal measurements (Table 1) in the patients during the pre-surgical diagnostic stage (T1) and at post-surgical (T2) follow-up after orthognathic surgery. The anatomical landmarks proposed by Vitral et al. [13] were used to measure the anterior joint space (AJS), the superior joint space (SJS), and the posterior joint space (PJS) in the sagittal direction (Figure 1).

#### 2.1.2. TMJ Measurement

Two strategies were used to perform the measurements to obtain information about the joint space and the location of the condyle in the fossa.

(1)Joint space (JS):

The methodology proposed by Fraga et al. [14] was used to calculate the ratio between the anterior and posterior joint spaces (A/P). It was calculated by dividing the anterior joint space by the posterior joint space, thereby obtaining the result of 1 mm (±0.09) as the centered condylar position. An A/P ratio greater than 1 mm was considered to indicate a posterior condylar position, and an A/P ratio less than 1 mm indicated a condylar shift that occurred anteriorly.

(2)Anteroposterior index (API):

The methodology proposed by Pullinger and Hollender was used [15] to define the position of the condyle in the temporal fossa using the following formula: API = (PJS − AJS)/(PJS + AJS) × 100. A position was considered to be centered when the API was between −12 and +12, an anterior position when the API was greater than +12, and a posterior position when the API was less than −12. The position displacement between the preoperative and postoperative condylar positions was classified as no change, posterior displacement, or anterior displacement. 

### 2.2. Statistical Analysis

Among the 26 cases, the same observer performed the measurements at different times within a two-week interval. An intraclass index of 0.74 was obtained for the continuous variables. Data analysis was performed using Graph Prism v. 9.5.1 (GraphPad by Domatics, Boston, MA, USA). The clinical parameters are shown as mean (X) and standard deviation (SD). The Shapiro–Wilk test was used for the analysis of normal distribution. To evaluate and compare the continuous variables before and after orthognathic surgery, a Student’s *t*-test was used, while considering a value of *p* < 0.05 as indicating a significant difference. 

## 3. Results

Fifty-two joints were analyzed pre- and postoperatively. The subjects ranged in age from 18 to 55 years (27.9 ± 10.81). Twelve were male (46.2%) and fourteen were female (53.8%). Fifteen subjects belonged to group CII and eleven subjects belonged to group CIII.

When comparing the measurements of the joint space in the sagittal position of patients in group CII at T1 and T2 (Table 2), we noted significant changes in the AJS (*p* = 0.0003), SJS (*p* = 0.0001), and PJS (*p* = 0.0001) relations. The AJS presented an average space of 1.41 mm at T1, whereas at T2, it reached an average distance of 1.79 mm, thus increasing by 0.38 mm. The SJP presented an average starting point of 2.72 mm at T1, whereas at T2, it had an average of 1.52 mm, showing a decrease of 1.2 mm. The PJS presented an average starting point of 2.92 mm at T1, whereas at T2, it had an average of 2.49 mm, thus decreasing by 0.43 mm. 

In relation to the measurements of the subjects in group CIII at the preoperative and postoperative stages (Table 3), significant changes were noted in all the measurements regarding the AJS (*p* = 0.0004), SJS (*p* = 0.003), and PJS (*p* = 0.02). For the AJS at T1, an average of 2.34 mm was observed, whereas at T2, there was an average distance of 1.74 mm, showing a decrease of 0.6 mm. The average distance of the PJS at T1 was 1.31 mm, whereas the distance at T2 was 1.54 mm, showing an increase of 0.23 mm.

When analyzing the sagittal position of the condyle in the articular cavity (Table 4), we did not find statistically significant differences in the change of the joint space (JS) related to the condylar position (group CII *p* < 0.18 and group CIII *p* < 0.25). In the same line, the anteroposterior index (API) of the condyle showed no differences between T1 and T2 (group CII *p* < 0.57 and group CIII *p* < 0.14). 

In group CII at T1, we observed 63.3% of the condyles in the anterior position based on the JS measurements and 56.6% based on the API evaluation. At T2, we observed 43% in the anterior position and 36.6% in the centered position based on the JS measurements. For API, both the percentages of the condyle in the anterior and centered positions were 43.3%.

In group CIII (Table 5) at T1, the JS and API measurements presented greater frequency of the condyle in the posterior position (JS: 68.2% and API: 59.1%). At T2, the JS measurements presented a greater frequency of the condyle in the centered position (45.6%), followed by the posterior position (40.9%). For API at T2, there was a greater frequency of the condyle in the centered position (50.0%), followed by the posterior position (40.9%). No statistical differences were observed between T1 and T2.

## 4. Discussion

Orthognathic surgery is considered a safe and stable procedure, with few surgical complications [16,17]. Although orthognathic surgery is not considered the primary treatment for managing or preventing TMJ disorder [18], it can reduce myofascial pain through anatomical corrections [19], improving the masticatory efficiency and balance of the stomatognathic system. This can reduce symptoms and variables associated with temporomandibular disorders [20]. Sahu et al. [21] performed diagnoses of TMJ disorders in 56 patients before surgery and at 6 months postoperatively, observing that the risk of developing a TMJ disorder or worsening of the joint conditions was very low.

Several authors [22,23,24] have mentioned that subjects with mandibular retrognathism present a greater prevalence of developing condylar resorption after surgery, resulting in a condylar remodeling process that is often associated with pain. On other hand, Podčernina et al. [25] evaluated the condylar position and volume of subjects with mandibular prognathism who underwent orthognathic surgery, and they did not detect any radiological signs of degeneration or bone remodeling nor significant changes in the condylar volume at one-year follow-up, regardless of the millimeters of mandibular movement.

Our results show that in the case of mandibular retrognathism treated with mandibular advancement surgery, the condyle tends to move away from the articular eminence and increase its anterior joint space; statistical differences were observed in the joint space. On the other hand, in the case of mandibula prognathism showing a posterior condylar position in the preoperative stage and after mandibular setback, the condyle tends to move away from the retrodiscal zone to increase the posterior joint space.

The condylar position into the fossa is related to the skeletal class [26], and surgical treatment with orthognathic surgery will be employed to correct the final position of the condylar head [27] as well as to reduce TMJ pain [28]. Ploder et al. [29] performed a study involving 375 subjects treated with orthognathic surgery; they showed significant changes in symptoms of TMJ disease after 2 years of follow up. In class II female patients with counter-clockwise maxillo-mandibular rotation, the authors observed complications related to TMJ symptoms during the first 6 weeks; however, this was absent at the 2-year follow up. In the same line, Kuhlefel et al. [30] performed a study involving 40 patients with class II skeletal deformity with symptoms of TMJ disarrangement; in 24 subjects, there were no observed changes in TMJ symptoms, whereas in 4 subjects, there were observed changes in TMJ symptoms. Yoon et al. [31], on other hand, performed a study involving 32 patients with class III skeletal deformity and TMJ disarrangements symptoms; they concluded that there was an absence of symptoms after 6 months.

Ashghpour et al. [32] performed orthognathic surgery in subjects with mandibular retrognathism with mandibular advancement movements smaller than 5 mm, finding an average of 0.32 mm in the anterior displacement of the disc, which might be associated with changes in the position of the condyle and its relation to the fossa, as observed in our study. However, our study did not perform an evaluation of the position of the joint disc, which is one of the weaknesses of our study.

Kim et al. [33] studied the condylar position of 33 subjects with mandibular prognathism before and after orthognathic surgery, and observed that, after 3 months of follow-up after mandibular setback, there was a greater prevalence of an anterior displacement of the condyle, which is in line with our results. 

Using the JS and API analyses, no statistically significant changes were observed; subjects with mandibular retrognathism exhibited a preferentially forward or centered position, and subjects with mandibular prognathism exhibited a preferentially centered or retruded position. The absence of a statistical correlation may be due to the normality range used to assess the condylar position, where the API has a wider normality range than the JS measurement. Another cause may be the small number of subjects in this study. The JS and API measurements evaluated the condylar position, showing that no differences were observed in terms of the position into the fossa; however, the joint space changed with an increase in some areas, demonstrating a movement of the condyle with no impact in terms of the tridimensional position in the fossa, thus maintaining the spatial position. 

Although we did not evaluate symptoms in this study, several authors [34,35,36] agree that the condylar position is not a parameter to determine joint pathologies; it has been demonstrated that symptomatic and asymptomatic joints can be in an anterior or a centered position. In the study by Al-Rawi et al. [36] using CBCT, the condylar position was evaluated in 35 patients without and 35 patients with a TMJ disorder, and the authors concluded that the condylar eccentricity by itself is not a sufficient variable for a diagnosis of a TMJ disorder since between 30% and 50% of the subjects with no joint pathology presented a non-centered position; however, the more tilted articular eminence was presented in a high frequency in subjects with symptoms. 

Manfredini et al. [35] indicated that, more frequently, skeletal profiles associated with mandibular retrognathism and hyperdivergent growth patterns present joint disc displacements and degenerative TMJ disorders. Lin et al. [37] indicated that women with mandibular retrognathism who presented greater angulation in their ANB measurement and mandibular plane more frequently had joint problems. The change in the position of the joint surface obtained after an orthognathic surgery must be considered during postoperative follow-up of patients. 

Bone changes in the condylar head after an orthognathic surgery is an adaptative reply to the mechanical load and stress in the TMJ with changes in the position of the condyle; in some cases, this can develop a pathological condition [38]. Condylar resorption is a clinical condition with a prevalence between 1% to 31%, which is mainly presented in female patients with class II and a high mandibular plane [39]. Catherine et al. [40] showed that these variables, together with estrogen deficiencies, a titled condylar head, and a mandibular movement over 10 mm, can increase the chance of condylar remodeling. In an observational study involving 15 subjects, Firoozei et al. [41] performed magnetic nuclear resonance 1 day before and 3 months after orthognathic surgery and found a disc position of 5.74 ± 1.21 at the preoperative stage and 5.65 ± 1.06 at the postoperative stage; they concluded that there were no changes in the condyle-to-disc ratio in short-term follow-up.

Because the aim of this research was to determine the preoperative and postoperative positions of the mandibular condyle in patients who underwent orthognathic surgery, no assessment was conducted on disk position. The methodology used in this research with CBCT can be applied to confirm condylar morphology, joint morphology, and the position of the condyle in the articular fossa [7].

Dalili et al. [42] performed measurements in the joint space as we did in our research, finding a centered position of the condyle in 92.5% of subjects with class I dental conditions. On the other hand, Imanimoghaddam et al. [43] reported a higher anterior joint space in subjects with TMJ disarrangements. Al-Rawi et al. [36] used another method of measurements and found differences between affected TMJ and healthy TMJ; in the case of female subjects with a TMJ disorder, disarrangement was observed in the bigger areas in the joint space when compared to healthy subjects. The use of CBCT in TMJ analysis can provide knowledge on the morphology of the hard structure and the joint space in the posterior, anterior, or upper space, as we presented in this study. 

The limitation of this study are related to the absence of disk assessment and the lack of an analysis of the defined signs or symptoms of TMJ pathology. Because of the lack of magnetic resonance imaging, a complete diagnosis of TMJ pathology could not be performed. However, the clinical significance of this study lies in its establishment of the final position of the condyle in terms of the movement related to sagittal facial deformity. These findings can be included in diagnosis and 3D planning to address the potential movement of the condyle, the potential impact on the final position of the mandible, and the relapse of the movement.

## 5. Conclusions

Based on the findings, we can conclude that orthognathic surgery causes changes in the sagittal position of the mandibular condyle in subjects with mandibular retrognathism and prognathism. Long-term studies are needed to determine the long-term condition of the TMJ; an analysis of the position of the joint disc and related symptoms should be included.

## Figures and Tables

**Figure 1 jpm-13-01544-f001:**
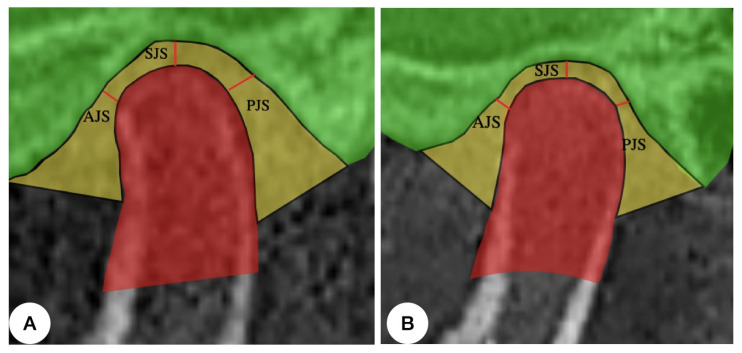
TMJ sagittal view of the anterior joint space (AJS), superior joint space (SJS), and posterior joint space (PJS) during the pre-surgical diagnostic stage (**A**) and at post-surgical (**B**) follow-up after orthognathic surgery.

**Table 1 jpm-13-01544-t001:** Anatomical landmarks of the temporomandibular joint in the sagittal and coronal directions used in this study.

Landmark	Definition
AJS	Distance between the most anterior point of the condylar surface and the posterior wall of the articular tubercle.
SJS	Distance between the superior point of the condylar surface and the most superior point of the articular fossa.
PJS	Distance between the posterior point of the condylar surface and the posterior wall of the articular fossa.
JS	Joint space related to the anterior and posterior measurements of the condyle in the articular fossa.
API	Position of the condyle in the temporal fossa using anterior and posterior measurements according to the formula proposed by Pullinger and Hollender.

**Table 2 jpm-13-01544-t002:** Measurement of the anterior, superior, and posterior joint spaces of the condyle at the preoperative and postoperative stages among patients in group CII.

	T1				T2				
	Right TMJ		Left TMJ		Right TMJ		Left TMJ		
	X	SD	X	SD	X	SD	X	SD	*p* < 0.05
AJS	1.34 mm	0.82	1.48 mm	1.14	1.74 mm	0.45	1.84 mm	0.56	0.0003 *
SJS	2.61 mm	1.36	2.83 mm	0.71	1.51 mm	0.65	1.52 mm	0.77	0.0001 *
PJS	3.03 mm	1.58	2.82 mm	1.31	2.61 mm	0.71	2.37 mm	0.64	0.0001 *

Note: CII: skeletal class II. T1: preoperative stage. T2: postoperative stage. AJS: anterior joint space. SJS: superior joint space. PJS: posterior joint space. X: average of measurements. SD: standard deviation. (*) indicates a statistically significant difference.

**Table 3 jpm-13-01544-t003:** Measurement of the anterior, superior, and posterior joint spaces of the condyle at the preoperative and postoperative stages among patients in group CIII.

	T1				T2				
	Right TMJ		Left TMJ		Right TMJ		Left TMJ		
	X	SD	X	SD	X	SD	X	SD	*p* < 0.05
AJS	2.53 mm	1.2	2.15 mm	0.99	1.81 mm	0.87	1.68 mm	0.63	0.0004 *
SJS	2.31 mm	0.63	1.98 mm	0.89	1.55 mm	0.59	1.82 mm	1	0.003 *
PJS	1.45 mm	0.58	1.17 mm	0.73	1.63 mm	0.57	1.46 mm	0.51	0.02 *

Note: CIII: skeletal class III. T1: diagnostic stage. T2: postoperative stage. AJS: anterior joint space. SJS: superior joint space. PJS: posterior joint space. X: average of measurements. SD: standard deviation. (*) indicates a statistically significant difference.

**Table 4 jpm-13-01544-t004:** Distribution of the sagittal position of the mandibular condyle in the joint space among subjects in group CII at the preoperative and postoperative stages.

		T1		T2		
		N	%	N	%	*p* < 0.05
JS	Centered	2	6.6	11	36.6	0.18
	Anterior	19	63.3	13	43.3	
	Posterior	9	30.1	6	20.1	
API	Centered	6	20	13	43.3	0.57
	Anterior	17	56.6	13	43.3	
	Posterior	7	23.3	4	13.3	

Note: CII: group CII. JS: joint space. API: anteroposterior index. N: number of joints analyzed.

**Table 5 jpm-13-01544-t005:** Distribution of the sagittal position of the mandibular condyle in the joint space in CIII group at the diagnostic and postoperative stages.

		T1		T2		
		N	%	N	%	*p* < 0.05
JS	Centered	4	18.1	10	45.4	0.25
	Anterior	3	13.6	3	13.6	
	Posterior	15	68.2	9	40.9	
API	Centered	6	27.2	11	50	0.14
	Anterior	3	13.6	2	9.1	
	Posterior	13	59.1	9	40.9	

Note: CIII: group CIII. JS: joint space. API: anteroposterior index. N: number of joints analyzed.

## Data Availability

The data are available from the corresponding author upon request.

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
