# Peer review of "Condylar Positional Changes in Skeletal Class II and Class III Malocclusions after Bimaxillary Orthognathic Surgery"

_jpm, 2023, doi:10.3390/jpm13111544_

Round 1
Reviewer 1 Report
Dear. Author, This manuscript is well organized, has no significant flaws in logical development, and reads well. However, there are a few corrections that need to be made. 1. On lines 94 and 95, the author describes the position of the patients and the state of their lips when the images were acquired. Were the patients' teeth occluded or in a physiologic rest position when the images were taken? 2. Verify that the units are appropriate for the anterior and posterior joint space ratio described on line 117. 3. It would also be good to add the authors' thoughts on the clinical significance of the findings presented in this study. Thank you.Author Response
On lines 94 and 95, the author describes the position of the patients and the state of their lips when the images were acquired. Were the patients' teeth occluded or in a physiologic rest position when the images were taken?
R: the position of the patient was in a physiological position
Verify that the units are appropriate for the anterior and posterior joint space ratio described on line 117.
R: The units are fine. Thank you
It would also be good to add the authors' thoughts on the clinical significance of the findings presented in this study. Thank you.
R: The paragraph was included in the discussion
Reviewer 2 Report
This clinical study provides meaningful information about the condylar positional changes after double jaw surgery. I have some suggestions for the authors as following.
It is easier to evaluate the patients of class 2 or class 3 as the article mentioned. However, most of the patients would have asymmetric deformities clinically and the condylar position after orthognathic correction may be different form one side to the other. I hope authors should give the detail description about the data of unilateral condylar changes separately.
The authors reported the postoperative results of condylar position. Is there any relationship between the occlusion stability and the condylar position? So the information about the patients' occlusion after operation should be added.
Acceptable.
Line 88, 21 days?
Line 229, after of?
Line 240, fosse
Author Response
It is easier to evaluate the patients of class 2 or class 3 as the article mentioned. However, most of the patients would have asymmetric deformities clinically and the condylar position after orthognathic correction may be different form one side to the other. I hope authors should give the detail description about the data of unilateral condylar changes separately.
R: Thank you: The facial asymmetry is assessed in another research. In this paper no asymmetry was included, evaluated clinically and in the CBCT; the reason to exclude this type of patients was the same as provided by the reviewer. Because the quantity of data to be included in a unique article, the authors designed a new article to show the new data as the maxilla mandibular movement and the unilateral change in a symmetric and non-asymmetric subjects.
The authors reported the postoperative results of condylar position. Is there any relationship between the occlusion stability and the condylar position? So the information about the patients' occlusion after operation should be added.
R: Thank you: Because the time involved in this research (6 mo. follow-up) was no possible to include the final occlusion in the research; however, the limitations and clinical significance included in the new version of our research showed the potential involvement in the final dental occlusion.
Reviewer 3 Report
Thank you for the opportunity to review interesting your article.
This study evaluated the condylar positional change in class II and II malocclusions.
The result is that a significant change in the anterior, superior and posterior space in both groups. The conclusion is that the sagittal position of the condyle is changed during orthognathic surgery.
The limitation of this study is not performed disk position analysis with MR.
The symptom of TMJ like pain, mouth opening withs, clicking etc., was not evaluated this study. The novel finding from this study is not found.
The discussion is not well constructed, and the result of this study was not discussed with references.
I have some questions about your article.
Introduction
L38- 41. Is this paragraph needed?
Material and Methods
Statistical analysis: add the company and production of Graph Prism.
Result
L142-146:It is easy to understand to include the table 2.
L159-162: It is easy to understand to include the table 3.
L190: Change Table V to 5.
Discussion
L249-250: This sentence is difference from conclusion.
Reference
L344-345: Missed the published years and pages.
Moderate English editing required.
Author Response
Thank you for the opportunity to review interesting your article. This study evaluated the condylar positional change in class II and II malocclusions. The result is that a significant change in the anterior, superior and posterior space in both groups. The conclusion is that the sagittal position of the condyle is changed during orthognathic surgery.
R: Thank you
The limitation of this study is not performed disk position analysis with MR. The symptom of TMJ like pain, mouth opening withs, clicking etc., was not evaluated this study.
R: The limitation was included in the article
The novel finding from this study is not found.
R: The finding of this article is related to position of the condyle using 3D analysis with CBCT and comparison with another research was performed.
The discussion is not well constructed, and the result of this study was not discussed with references.
R: Discussion was modified
I have some questions about your article.1. Introduction L38- 41. Is this paragraph needed?
R: Thank you; in opinion of the authors, this paragraphs can help in the introduction of the problem related to Condyle position.
Material and Methods: Statistical analysis: add the company and production of Graph Prism.
R: Included, thank you
L142-146:It is easy to understand to include the table 2.
R: Thank you
L159-162: It is easy to understand to include the table 3.
R: Thank you
L190: Change Table V to 5.
R: the modified was realized.
Discussion
L249-250: This sentence is difference from conclusion.
R: the sentence was modified
Reference
L344-345: Missed the published years and pages.
R: the reference was modified
Reviewer 4 Report
Dear Authors,
Thank you for your manuscrip.
First of all, i would like if you have ethic committe for doing cbct before and after orthognatic surgery. I really do not understand the usefulness of doing that.
I think there is not the power of the study.
It is not clear if those patients have any signs and symptoms of TMD. Can you explain it? I would have assessd that before doing the study.
Furthermore, what are the reason of surgery? just to correct the skeletal class or there are malocclusions ( dental ones)?
I would suggest to clarify the kind of change of condylar position in relation to the kind of class. i do not think in the second class it would be the same of the third.
You did not clarify if these movements of the condyle are directed to a healthy position or not.
I do not understand the scientific novelty in this study, i really think it is very obvious that condyle change position. it would have been interesting to lknow if signs and symptoms improve and how much it moves and the changes during the following months. It is not even necessary CBCT, but just an x ray to avoid eccessive exposure.
it is quite acceptable.
Author Response
- First of all, i would like if you have ethic committe for doing cbct before and after orthognatic surgery. I really do not understand the usefulness of doing that.
- R: Dear reviewer. Yes, we have EC included in the paper. On other hand, we do not understand the comment because the CBCT is used in regular clinical practice of maxillofacial surgery; in our team we use for virtual planning and 3D print of the surgical splints in all the cases (not only in the subjects included in this research); for that reason, the use previously to surgery is fully indicate in orthognathic surgery. The 6 months follow-up is performed with CBCT in all of the patients in the department and the hospital; the reason is to evaluate the conditions of the patient in the surgical process. The last CBCT in our cases is using after the braces are removed. The irradiation included in CBCT is related to the size of the projection; however, is close to 68 µSv in the New Tome machine, 10 to 15 times less than CT scan. In comparison, lateral cephalogram use 5.5 µSv and the panoramic radiography use 3.85 μSv to 30 μSv and none of these are used in 3D planning. The CBCT is a good exam to perform 3D planning in maxillofacial surgery and is a good indication for follow up of the operated patients.
- I think there is not the power of the study. 3. It is not clear if those patients have any signs and symptoms of TMD. Can you explain it? I would have assessd that before doing the study.
- R: Thank you; the assessment of TMD was not used in this research because the aim of this paper was to know about the final position of the condyle and the changes related to orthognathic surgery; no MR was used in the protocol. For the aim of this research, the use of CBCT to perform a morphological analysis of the TMJ is a good choice.
- Furthermore, what are the reason of surgery? just to correct the skeletal class or there are malocclusions ( dental ones)?
- R: The reason of the surgery was related to obtain a stable dental occlusion and the treatment of the dentofacial deformity
- I would suggest to clarify the kind of change of condylar position in relation to the kind of class. i do not think in the second class it would be the same of the third.
- R: In the Material and method was presented the inclusion criteria related to the class II and class III dentofacial deformity; class I was not included. Tables 2 to 5 show class II or class III conditions and the results of each group.
- You did not clarify if these movements of the condyle are directed to a healthy position or not.
- R: was not evaluated the “healthy” condition of the condyle or TMJ; the aim was to know the position of the condyle into the fossa; maybe, in the future we can conduct a research looking a treatment of TMD and the “healthy” position would be evaluate.
- I do not understand the scientific novelty in this study, i really think it is very obvious that condyle change position. it would have been interesting to lknow if signs and symptoms improve and how much it moves and the changes during the following months. It is not even necessary CBCT, but just an x ray to avoid excessive exposure.
- R: Dear reviewer; thank you for your comment, however, we are note in agree with you. X Ray cannot provide good and reproductive results in term of TMJ analysis.We are using for a long time CBCT to assess the formal treatment of dentofacial deformity and the follow-up of the patients; Exposure and radiation are not a problem. The irradiation included in CBCT is related to the size of the projection; however, is close to 68 µSv in the New Tome machine, 10 to 15 times less than CT scan. In comparison, lateral cephalogram use 5.5 µSv and the panoramic radiography use 3.85 μSv to 30 μSv.
Round 2
Reviewer 3 Report
Thank you for reviewing revised manuscript. Unfortunately, this article has no novel finding. The discussion was not well constructed after revision.
There are some spell and grammatical mistakes.
I recommend the native speaker check again.
Author Response
no comments
Reviewer 4 Report
i think you could correct it better,
Author Response
no comments